# Inverse modeling, analysis and control of twin rotor aerodynamic systems with optimized artificial intelligent controllers

Ahmad Al-Talabi[1]*, Taqwa Oday Fahad[2], Aqeel Abdulazeez Mohammed[3],
Ali Hussien Mary[4]

**1** Department of Medical Instrumentation Techniques, College of Engineering and Information Technology, AlShaab University, Baghdad, Iraq, **2** Biomedical Engineering Department, University of Technology, Baghdad, Iraq, **3** Electronic and Communications Engineering, College of Engineering, University of Baghdad, Baghdad, Iraq, **4** Mechatronics Engineering Department, Al-Khwarizmi College of Engineering, University of Baghdad, Baghdad, Iraq

* ahmad.altalabi@alshaab.edu.iq

## Abstract

This paper suggests a novel optimal inverse Radial Basis Function (RBF) neural network model for the control of Twin Rotor Aerodynamic Systems (TRAS), such as Multi-Input–Multi-Output (MIMO) systems with high nonlinearity and coupling effects between channels. After analyzing and linearizing the dynamic model, TRAS is decoupled into two Single Input Single Output (SISO) systems, thereby creating vertical (pitch model) and horizontal (yaw model) systems. The relationship between the output angle of each subsystem and the input voltage is modeled using the inverse RBF neural network. The weights, biases, centers and widths of the Gaussian function are unknown parameters of the proposed inverse neural model, and they are obtained using Atom Search Optimization (ASO). A combination of the proportional derivative controller and the proposed inverse neural model fed forward controller is then applied to control the angles of each subsystem with different conditions. The simulation results showed that the proposed controller demonstrates noticeable performance improvements over the Fractional Order PID (FOPID) and Particle Swarm Optimization-PID (PSO-PID) controllers. Compared to FOPID, it achieves an 88.3% faster rise time, a 96.0% faster settling time, and a 93.8% lower overshoot for the Yaw model, along with a 42.8% faster rise time, a 73.9% faster settling time, and an 86.8% lower overshoot for the Pitch model. In comparison to PSO-PID, the Yaw model shows a 36.2% faster rise time, an 86.7% faster settling time, and a 59.7% lower overshoot, while the Pitch model exhibits a 58.4% slower rise time but compensates with a 59.9% faster settling time and a 71.2% lower overshoot. Additionally, integral performance indices are notably reduced for the proposed controller.

**Data availability statement:** Data relevant to this study are available from Dryad at https://doi.org/10.5061/dryad.z08kprrqw.

**Funding:** The author(s) received no specific funding for this work.

**Competing interests:** The authors have declared that no competing interests exist.

## 1. Introduction

A Twin Rotor Aerodynamic System (TRAS) is a laboratory model that can be considered a helicopter prototype [1,2]. TRAS is a multi-input multi-output system with two degrees of freedom according to the angles that determine its posture, which are known as the pitch and yaw angles. Moreover, it is a nonlinear system with high coupling effects between channels [3]. Thus, controlling TRAS is challenging, and has motivated many researchers to develop other appropriate controllers [4–13]. In [13], the authors proposed a flexible mixed-optimization framework with $H_\infty$ control for a twin rotor MIMO system, combining Nonlinear Dynamic Inversion (NDI), to linearize the system, and a mixed optimization control technique based on the Method of Inequality (MOI). This work highlights the effectiveness of integrating advanced control strategies to manage complex, nonlinear systems with significant coupling effects. PID controller is an essential controller widely used to control many systems due to its ease of implementation [14]. However, it often has difficulty meeting the performance demands of high nonlinearity systems like TRAS [15]. To mitigate this limitation, numerous control schemes propose hybrid approaches that integrate a PID controller with an intelligent control scheme [16]. In [17] PID is combined with a fuzzy compensator to track desired attitudes quickly, and all parameters are tuned using a Real-value Genetic Algorithm (RGA). In [18], the authors proposed Fractional Order PID controllers (FOPID) and tuning the parameters using Particle Swarm Optimization (PSO). Analysis of the results has shown that FOPID is outperformed and smoother than the traditional PID controllers [19]. The Sugeno Fuzzy PD-like System (SFPDS) is introduced to improve the transient and steady-state response of TRAS [20]. Moreover, sliding mode control is an efficient robust control approach that has been used successfully for controlling nonlinear systems [21–23]. In [24], the authors presented an adaptive sliding mode controller to control TRAS. The Lyapunov theorem has been used to derive an adaptive control law that compensates for uncertainty and reduces chattering in the control signal. Recently, considerable effort has been made to present different methods to identify and control nonlinear systems with uncertain dynamics. In this line, [25] identified linear and nonlinear systems based on input and output signals. In [26], the coefficients of rotorcraft aerodynamics are estimated by the Radial Basis Function (RBF) neural network. In [27], the Chebyshev Neural Network (CNN) was used as an observer of TRAS.

This paper investigates the optimal inverse model to control TRAS with an emphasis on explaining the inverse optimal model control scheme and its relevance to TRAS. It also proposes an inverse RBF neural network model based on the Atom Search Optimization (ASO) algorithm. The inverse RBF neural network represents the main component of the proposed controller. Unlike traditional PID controllers, which rely on fixed gains and linear control laws, the inverse RBF neural network can adapt to system nonlinearities. Specifically, it can approximate complex nonlinear functions, allowing it to model and compensate for high system nonlinearities that traditional PID controllers cannot effectively handle. The network has the ability to learn and adjust its parameters to ensure optimal performance, even in the presence

of uncertainties or changes in operating conditions. Accordingly, the ASO algorithm, as an optimization algorithm, is used to optimize the parameters of the inverse RBF neural network. As a result, this algorithm enhances the controller's performance because the ASO algorithm acts as a global optimizer to find the optimal parameters. Moreover, it dynamically adjusts the balance between exploration (searching for new solutions) and exploitation (refining existing solutions) to ensure robust and efficient optimization. To improve the effectiveness of the proposed optimal inverse neural model under TRAS control, a novel hybrid controller that combines a proportional derivative with inverse RBF neural model, optimized by the ASO algorithm, was proposed for controlling TRAS to track different trajectories. This hybrid approach exploits the stability and simplicity of the PD controller while utilizing the adaptive and learning capabilities of the inverse RBF to handle system nonlinearities and uncertainties. To the best of our knowledge, this specific combination has not been extensively explored in the context of TRAS. The main contributions of this work are:

(1) ASO, which was firstly proposed by [28], is considered a new base for inverse neural modelling, which was first presented to design the inverse model of TRAS.

(2) To reduce the computational time in the training stage, the TRAS is linearized and decoupled into two Single Input Single Output (SISO) subsystems.

(3) The modeling error was cancelled by hybridizing the inverse neural controller with the PD control term that acts as a correction error term.

(4) Bound Input Bounded Output (BIBO) stability was approved for the inverse RBF neural network controller.

(5) The stability of the closed loop system was approved based on the Lyapunov stability theorem.

(6) The presented controller is model-free, in which there is no need to determine the dynamic model of TRAS.

The remainder of this paper is structured as follows: The dynamic model of a TRAS is presented in Section 2. In Section 3, we describe the linearized model of TRAS and how it can be decoupled into two SISO systems. A brief introduction to the Atom Search Optimization (ASO) algorithm is given in Section 4, whereas an overview of the RBF neural network structure is provided in Section 5. Section 6 focuses on describing the inverse neural modeling of TRAS, while Section 7 is dedicated to the implementation of the ASO algorithm for inverse modeling. Section 8 addresses the determination of the unknown parameters of the inverse model. Section 9 discusses the proposed control model, while Section 10 provides the closed-loop stability analysis of the system using the Lyapunov stability theorem. Section 11 presents simulations of the system identified in the previous sections to demonstrate the feasibility of the proposed controller. Finally, the paper is concluded in Section 12.

## 2. Dynamic model of a TRAS

The Twin Rotor Aerodynamic System is a laboratory setup that resembles a helicopter system, as shown in Fig 1. There are two rotors, known as the main and the tail. The main rotor generates vertical thrust enabling the system to rotate in the vertical plane around the horizontal Z and X axes. The tail rotor generates horizontal thrust, which enables the system to rotate in the horizontal plane around the vertical. The voltages applied to these rotors represent the control inputs. The beam deviations in the horizontal plane and vertical plane are called azimuth and pitch angles, respectively, and these angles are measured outputs of the system.

In [18], the authors provide a detailed derivation of the dynamic model based on Lagrange's equation. The following differential equations can represent the nonlinear dynamic model:

$$\ddot{\varnothing} = \frac{1}{J\cos^2\theta + J_A}(F_\varnothing l_t cos\theta - c_\varnothing \dot{\varnothing} - k_\varnothing \varnothing + J\dot{\varnothing}\dot{\theta}sin2\theta)$$

(1)

$$\ddot{\theta} = \frac{1}{J_\theta}(F_\theta l_m - k_1 cos\theta - k_2 cos\theta - c_\theta \dot{\theta} - k_\theta \theta - \frac{J}{2}\dot{\varnothing}^2 sin2\theta)$$

(2)

$$\dot{w}_\varnothing = -\frac{1}{T_\varnothing}w_\varnothing + \frac{g_\varnothing}{T_\varnothing}u_\varnothing$$

(3)

$$\dot{w}_\theta = -\frac{1}{T_\theta}w_\theta + \frac{g_\theta}{T_\theta}u_\theta$$

(4)

where $\varnothing$ is the yaw angle, $\theta$ is the pitch angle, $w_\varnothing$ is the angular velocity of the tail rotor, $w_\theta$ is the angular velocity of the main rotor, $u_\varnothing$ is the input voltage to the tail rotor, $u_\theta$ is the input voltage to the main rotor, $l_m$ is the length of the main part of the beam and $l_t$ is the length of the tail part of the beam. Table 1 lists the nominal system parameters. Defining $x$ as a state vector and $y$ as an output vector:

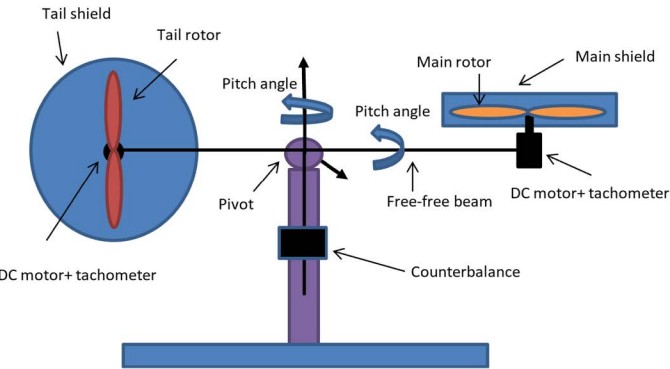

**Fig 1. The laboratory setup of TRAS.**

**Table 1. TRAS parameters.**

| Parameter | Description | Value | Units |
|---|---|---|---|
| $J_A$ | Moment of inertia of the vertical rotor | 0.0561 | kg. m² |
| $J$ | Moment of inertia of the horizontal rotor | 0.2168 | kg. m² |
| $J_\theta$ | Moment of inertia of the beam about the pitch axis | 0.0559 | kg. m² |
| $m_m$ | Mass of the main part of the beam | 0.0145 | kg |
| $m_t$ | Mass of the tail part of the beam | 0.0155 | kg |
| $l_m$ | Length of the main part of the beam | 0.2400 | m |
| $l_t$ | Length of the tail part of the beam | 0.2500 | m |
| $c_\theta$ | Damping coefficient of the pitch axis | 0.010 | Nm s/rad |
| $c_\varnothing$ | Damping coefficient of the yaw axis | 0.010 | Nm s/rad |
| $k_\theta$ | Stiffness coefficient of the pitch axis | 0.06 | rad/s |
| $k_\varnothing$ | Stiffness coefficient of the yaw axis | 0.06 | rad/s |
| $k_1$ | Torque constant of the vertical rotor | $5.00 \times 10^{-2}$ | Nm |
| $k_2$ | Torque constant of the horizontal rotor | $9.36 \times 10^{-2}$ | Nm |
| $g_\theta$ | Gear ratio of the pitch axis | 22.7273 | rad/s |
| $g_\varnothing$ | Gear ratio of the yaw axis | 18.1818 | rad/s |
| $T_\theta$ | Torque constant of the pitch axis | 2.5 | kg.m². Rad/s |
| $T_\varnothing$ | Torque constant of the yaw axis | 5 | kg.m². Rad/s |

$$x = \begin{bmatrix} \varnothing & \dot{\varnothing} & w_\varnothing & \theta & \dot{\theta} & w_\theta \end{bmatrix}^T \tag{5}$$

$$y = \begin{bmatrix} \varnothing & \theta \end{bmatrix}^T \tag{6}$$

The state space model can be expressed as:

$$\dot{x} = f(x, t) = \begin{bmatrix} f_1 \\ f_2 \\ f_3 \\ f_4 \\ f_5 \\ f_6 \end{bmatrix} \tag{7}$$

$$= \begin{bmatrix} x_2 \\ \frac{1}{Jcos^2x_4 + J_A}(F_1 l_t \cos x_4 - c_\varnothing x_2 \\ -k_\varnothing x_1 + Jx_2 x_5 \sin 2x_4) \\ -\frac{1}{T_\varnothing}x_3 + \frac{g_\varnothing}{T_\varnothing}u_\varnothing \\ x_5 \\ \frac{1}{J_\theta}(F_2 l_m - c_\theta x_5 - k_\theta x_4 - k_1 \cos x_4 \\ -k_2 \sin x_4 - Jx_2^2 \sin 2x_4) \\ -\frac{1}{T_\theta}x_6 + \frac{g_\theta}{T_\theta}u_\theta \end{bmatrix} \tag{8}$$

$$y = \begin{bmatrix} 1 & 0 & 0 & 0 & 0 & 0 \\ 0 & 1 & 0 & 0 & 0 & 0 \end{bmatrix} x \tag{9}$$

## 3. Linearization and decoupling of the TRAS model

The linearized model of TRAS can be derived using the Jacobian Linearization technique around the operating point $(x_0, u_0)$.

$$\dot{x} = Ax + Bu \tag{10}$$

$$A = \left.\frac{\partial f}{\partial x}\right|_{(x_0, u_0)}, B = \left.\frac{\partial f}{\partial u}\right|_{(x_0, u_0)}$$

$$A = \begin{bmatrix} \frac{\partial f_1}{\partial x_1} & \frac{\partial f_1}{\partial x_2} & \frac{\partial f_1}{\partial x_3} & \frac{\partial f_1}{\partial x_4} & \frac{\partial f_1}{\partial x_5} & \frac{\partial f_1}{\partial x_6} \\ \frac{\partial f_2}{\partial x_1} & \frac{\partial f_2}{\partial x_2} & \frac{\partial f_2}{\partial x_3} & \frac{\partial f_2}{\partial x_4} & \frac{\partial f_2}{\partial x_5} & \frac{\partial f_2}{\partial x_6} \\ \frac{\partial f_3}{\partial x_1} & \frac{\partial f_3}{\partial x_2} & \frac{\partial f_3}{\partial x_3} & \frac{\partial f_3}{\partial x_4} & \frac{\partial f_3}{\partial x_5} & \frac{\partial f_3}{\partial x_6} \\ \frac{\partial f_4}{\partial x_1} & \frac{\partial f_4}{\partial x_2} & \frac{\partial f_4}{\partial x_3} & \frac{\partial f_4}{\partial x_4} & \frac{\partial f_4}{\partial x_5} & \frac{\partial f_4}{\partial x_6} \\ \frac{\partial f_5}{\partial x_1} & \frac{\partial f_5}{\partial x_2} & \frac{\partial f_5}{\partial x_3} & \frac{\partial f_5}{\partial x_4} & \frac{\partial f_5}{\partial x_5} & \frac{\partial f_5}{\partial x_6} \\ \frac{\partial f_6}{\partial x_1} & \frac{\partial f_6}{\partial x_2} & \frac{\partial f_6}{\partial x_3} & \frac{\partial f_6}{\partial x_4} & \frac{\partial f_6}{\partial x_5} & \frac{\partial f_6}{\partial x_6} \end{bmatrix}_{(x_0; u_0)} \tag{11}$$

$$B = \begin{bmatrix} \frac{\partial f_1}{\partial u_1} & \frac{\partial f_1}{\partial u_2} \\ \frac{\partial f_2}{\partial u_1} & \frac{\partial f_2}{\partial u_2} \\ \frac{\partial f_3}{\partial u_1} & \frac{\partial f_3}{\partial u_2} \\ \frac{\partial f_4}{\partial u_1} & \frac{\partial f_4}{\partial u_2} \\ \frac{\partial f_5}{\partial u_1} & \frac{\partial f_5}{\partial u_2} \\ \frac{\partial f_6}{\partial u_1} & \frac{\partial f_6}{\partial u_2} \end{bmatrix} \Bigg|_{(x_0; u_0)}$$

(12)

$$u = \begin{bmatrix} u_\varnothing & u_\theta \end{bmatrix}^T$$

(13)

By setting input voltages equal zero ($u_\varnothing = 0$; $u_\theta = 0$), and the derivatives of the yaw angle and pitch angle also equal zero ($\dot{\varnothing} = 0, \ddot{\varnothing} = 0, \dot{\theta} = 0, \ddot{\theta} = 0$), we can determine $\varnothing$ and $\theta$ from Equation (10), which are 0 and −0.7098 respectively. By substituting $x = [0\ 0\ 0\ {-}0.7098\ 0\ 0]$ and $u = \begin{bmatrix} 0 & 0 \end{bmatrix}$, the linearized TRAS model can be achieved.

$$A = \begin{bmatrix} A_1 & A_2 \\ A_3 & A_4 \end{bmatrix}$$

(14)

where:

$$A_1 = \begin{bmatrix} 0 & 1 & 0 \\ -0.3318 & -0.0553 & 0.1001 \\ 0 & 0 & -0.2000 \end{bmatrix}, A_2 = \begin{bmatrix} 0 & 0 & 0 \\ 0 & 0 & 0 \\ 0 & 0 & 0 \end{bmatrix}, A_3 = \begin{bmatrix} 0 & 0 & 0 \\ 0 & 0 & 0 \\ 0 & 0 & 0 \end{bmatrix}$$

$$A_4 = \begin{bmatrix} 0 & 1 & 0 \\ -2.9270 & -0.1789 & 3.4390 \\ 0 & 0 & -0.4000 \end{bmatrix}, B = \begin{bmatrix} 0 & 0 \\ 0 & 0 \\ 3.63 & 0 \\ 0 & 0 \\ 0 & 0 \\ 0 & 9.0909 \end{bmatrix}$$

(15)

Equation (14) indicates that TRAS can be decoupled into two SISO systems.

**For the yaw mode:**

$$\dot{x}_\varnothing = A_\varnothing x_\varnothing + B_\varnothing u_\varnothing$$

$$\dot{x}_\varnothing = \begin{bmatrix} \dot{\varnothing} \\ \ddot{\varnothing} \\ \dot{w}_\varnothing \end{bmatrix} = \begin{bmatrix} 0 & 1 & 0 \\ -0.3318 & -0.0553 & 0.1001 \\ 0 & 0 & -0.2000 \end{bmatrix}, \begin{bmatrix} \varnothing \\ \dot{\varnothing} \\ w_\varnothing \end{bmatrix} + \begin{bmatrix} 0 \\ 0 \\ 3.63 \end{bmatrix} u_\varnothing$$

(16)

**For the pitch model:**

$$\dot{x}_\theta = A_\theta x_\theta + B_\theta u_\theta$$

$$\dot{x}_\theta = \begin{bmatrix} \dot{\theta} \\ \ddot{\theta} \\ \dot{w}_\theta \end{bmatrix} = \begin{bmatrix} 0 & 1 & 0 \\ -2.9270 & -0.1789 & 3.4390 \\ 0 & 0 & -0.4000 \end{bmatrix}, \begin{bmatrix} \theta \\ \dot{\theta} \\ w_\theta \end{bmatrix} + \begin{bmatrix} 0 \\ 0 \\ 9.0909 \end{bmatrix} u_\theta$$

(17)

## 4. Atom search optimization

Atom Search Optimization (ASO) is a newly developed meta-heuristic optimization algorithm based on molecular dynamics, where each atom interacts with other atoms via interaction and constraint forces. ASO was firstly proposed in [28] and has proven its superiority over other optimization algorithms, including PSO [29]. In ASO, the atom position is updated by:

$$x_i^d(t+1) = x_i^d(t) + v_i^d(t+1) \tag{18}$$

where $x_i^d(t)$ and $x_i^d(t+1)$ are the positions of $i^{th}$ atom in the $d^{th}$ dimension at $t^{th}$ and $(t+1)^{th}$ iterations, respectively and $v_i^d(t+1)$ is the velocity of $i^{th}$ atom in the $d^{th}$ dimension at the $(t+1)^{th}$ iteration. This velocity is determined as follows:

$$v_i^d(t+1) = rand_i^d * v_i^d(t) + a_i^d(t) \tag{19}$$

where $rand$ is a random number lies between zero and one, and $a_i^d(t)$ is acceleration of the $i^{th}$ atom in the $d^{th}$ dimension at the $t^{th}$ iteration, and is calculated as follows:

$$a_i^d(t) = -\eta(t) \sum_{j \in Kbest} \frac{rand_j \left(2\left(h_{ij}(t)\right)^{13} - \left(h_{ij}(t)\right)^7\right)}{m_i(t)} \times \frac{x_i^d(t) - x_j^d(t)}{\left\| x_i(t), x_j(t) \right\|_2} + \beta e^{-\frac{20t}{T}} \frac{x_{best}^d(t) - x_i^d(t)}{m_i(t)} \tag{20}$$

where $Kbest$ is a subset consisting of the first $K$ atoms with the best fitness function values, $T$ is the maximum number of iterations, $x_{best}(t)$ is the best atom position in the $t^{th}$ iteration and $\beta$ is the weight of the Lagrangian multiplier. $m_i(t)$ represents the mass of the $i^{th}$ atom at the $t^{th}$ iteration, while $\eta(t)$ is the depth function used to adjust the repulsion or attraction region. It can be defined as follows:

$$\eta(t) = \alpha \left(1 - \frac{t-1}{T}\right)^3 e^{-\frac{20t}{T}} \tag{21}$$

where $\alpha$ is the depth weight. $h_{ij}$ is defined by:

$$h_{ij}(t) = \begin{cases} h_{min} & \frac{r_{ij}(t)}{\sigma(t)} < h_{min} \\ \frac{r_{ij}(t)}{\sigma(t)} & h_{min} \leq \frac{r_{ij}(t)}{\sigma(t)} \leq h_{max} \\ h_{max} & \frac{r_{ij}(t)}{\sigma(t)} > h_{max} \end{cases} \tag{22}$$

where $h_{min}$ and $h_{max}$ represent the lower and upper bounds of $h$, respectively, and $\sigma$ is the length scale representing the collision diameter, while $r_{ij}(t)$ is the distance between the $i^{th}$ and $j^{th}$ atoms at the $t^{th}$ iteration. $h_{min}$ and $h_{max}$ are calculated from:

$$h_{min} = g_0 + g(t) \tag{23}$$

$$h_{max} = u \tag{24}$$

with $g_0 = 1.1$, $u = 2.4$ and $g(t)$ is given by:

$$g(t) = 0.1 * sin(\frac{\pi}{2} \cdot \frac{t}{T}) \tag{25}$$

$m_i(t)$ can be calculated as follows:

$$m_i(t) = \frac{M_i(t)}{\sum_{j=1}^{N} M_j(t)}$$

(26)

$$M_i(t) = e^{-\frac{Fit_i(t) - Fit_{best}(t)}{Fit_{worst}(t) - Fit_{best}(t)}}$$

(27)

where $Fit_{worst}(t)$ and $Fit_{best}(t)$ refer to the atoms with the lowest and highest fitness values, respectively, at the $t^{th}$ iteration, while $Fit_i(t)$ represents the fitness value of the $i^{th}$ atom at the $t^{th}$ iteration. $\sigma(t)$ is defined by:

$$\sigma(t) = \left\| x_{ij}(t), \frac{\sum_{j \in Kbest} x_{ij}(t)}{K(t)} \right\|_2$$

(28)

where:

$$K(t) = N - (N-2)\sqrt{\frac{t}{T}}$$

(29)

$N$ represents the number of atoms.

## 5. Radial basis function neural network

A Radial Basis Function (RBF) neural network is a fed forward neural network containing only one hidden layer. The important features of RBF that enabled the researchers to exploit it are the structure simplicity, rapid training, and effective prediction accuracy [27]. Fig 2 illustrates the RBF architecture, assuming that there are $M$ input variables denoted by the vector $x = [x_1 \ x_2 \ldots\ldots.x_M]$. The first layer is the input layer, which feeds input values to the next layer; the hidden layer with $h$ nodes, and the Gaussian RBF is applied for each input.

The output of node $j$ in the hidden layer is:

$$\varnothing_j (\|x - \mu_j\|) = exp(\frac{\|x - \mu_j\|^2}{2b_j^2})$$

(30)

where $\mu_j$ and $b_j$ stand for the center point and width of the Gaussian function for node $j$, respectively. The final layer is the output layer, which determines the output of the RBF:

$$y = w_0 + \sum_{j=1}^{h} w_j \varnothing_j (\|x - \mu_j\|)$$

(31)

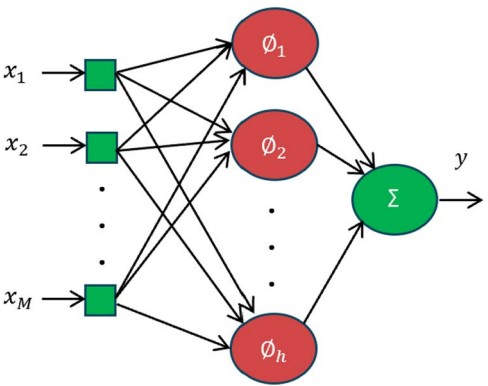

**Fig 2. RBF neural network structure.**

where $w_j$ for $j = 0, 1, 2, \ldots, h$ are the RBF neural network weights.

## 6. Inverse neural model identification of TRAS

The forward modeling of TRAS is used to calculate the yaw and pitch angles according to the input voltages $u_\varnothing$ and $u_\theta$. Furthermore, the inverse modeling of TRAS produces the required input voltages according to the given yaw and pitch angles. Generally, the inverse dynamic of any system can be written as:

$$u(t) = f(u(t-1), u(t-2), \ldots \ldots u(t-n), y(t-1), \ldots y(t-m)) \tag{32}$$

where $n$ and $m$ represent the maximum orders of the input and output, respectively, while $f$ is a nonlinear function represents the mapping of the RBF neural network in this work, as shown in Fig 3.

**For the yaw model:**

$$\hat{u}_{p\varnothing}(t) = IRBF_\varnothing(u_{p\varnothing}(t-1), u_{p\varnothing}(t-2), \ldots \ldots u_{p\varnothing}(t-n), \varnothing(t), \ldots \varnothing(t-m)) \tag{33}$$

$$\hat{u}_{p\varnothing} = w_{\varnothing 0} + \sum_{j=1}^{H} w_{\varnothing j} \varnothing_j \left( \left\| x_\varnothing - \mu_{\varnothing j} \right\| \right) \tag{34}$$

**For the pitch model:**

$$\hat{u}_{p\theta}(t) = IRBF_\theta(u_{p\theta}(t-1), u_{p\theta}(t-2), \ldots \ldots u_{p\theta}(t-n), \theta(t), \ldots \theta(t-m)) \tag{35}$$

$$\hat{u}_{p\theta} = w_{\theta 0} + \sum_{j=1}^{H} w_{\theta j} \varnothing_j \left( \left\| x_\theta - \mu_{\theta j} \right\| \right) \tag{36}$$

The weights, biases, centers, and widths of the Gaussian functions are unknown parameters that are determined in the identification process.

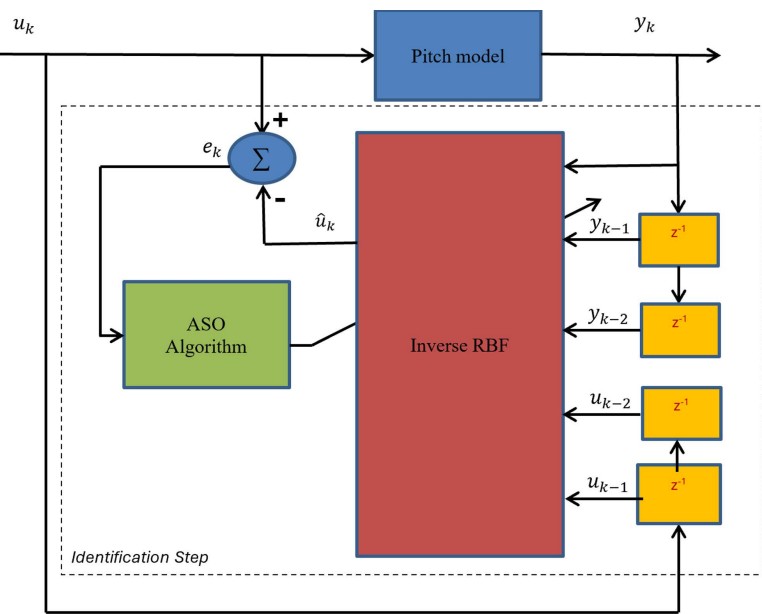

**Fig 3. Inverse RBF neural model.**

# 7. Tuning inverse RBF model by ASO

The main objective of the identification scheme is to get the optimal values for the unknown parameters of the proposed inverse RBF model of TRAS. To do so, the optimal parameters can be obtained by minimizing the differences between the real control signal and the predicted control signal. This is known as the objective function and is defined as:

$$Fit = \frac{1}{O} \sum_{r=1}^{O} \left( u(r) - \hat{u}(r) \right)^2 \tag{37}$$

where $u(r)$ is the output of the system, $\hat{u}(r)$ is the predicted control signal and $O$ is the number of elements in the training data. In this work, we utilize ASO to optimize the unknown parameters of the inverse RBF model, as follows:

Step 1: At first, the centers and widths of the Gaussian transfer function; weights between the hidden and output layers are coded to an atom population, where each atom is expressed by a real number.

Step 2: Define $m$ as the population size and create $m$ atoms with random values.

Step 3: Determine the objective function for each atom.

Step 4: Update the atoms according to the ASO module.

Step 5: Repeat Step 4 until the maximum number of iterations is reached.

The flow chart of the applied ASO algorithm to optimize the parameters of the inverse RBF model is depicted in Fig 4.

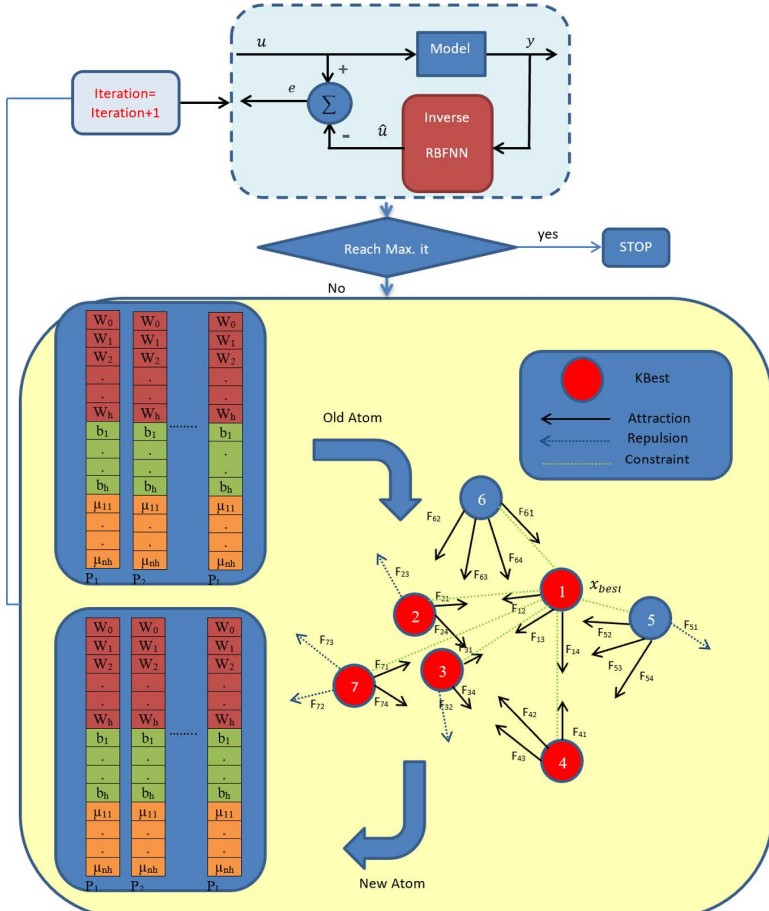

**Fig 4. The block diagram of the ASO algorithm implementation for inverse modeling.**

## 8. Identification results

This section is focused on the determination of the unknown parameters of the inverse RBF-ASO. A set of data is collected by applying square input voltages $u_\varnothing$ and $u_\theta$ to the pitch and yaw models and manipulating the output of each model $\varnothing$ and $\theta$. Figs 5 and 6 show the pitch and yaw angles according to the input signals. The proposed identification algorithm uses these data to optimize the parameters of the inverse RBF model. Before training the RBF for each model, the RBF structure is defined as shown in Fig 3. Four neurons in the hidden layer and 5 nodes in the input layer are defined as:

$$x = [y_k \; y_{k-1} \; y_{k-2} \; u_{k-1} \; u_{k-2}]$$

The training results and prediction error of the yaw and pitch models are given in Figs 5 and 6, respectively.

## 9. Controller design

The proposed inverse RBF-ASO–PD control model is discussed in this section. The block diagram of the proposed hybrid controller is shown in Fig 7. The inverse neural models have been trained offline to approximate the inverse dynamic of the pitch and yaw models of the linearized TRAS. In addition, the trained inverse models are integrated with the feedback PD controller to enhance the performance of the output angles of the TRAS.

The hybrid controller aims to overcome the inaccuracy that may result in the identification stage due to the modeling error and external disturbance. The proposed control law is:

$$u(t) = \hat{u}(t) + u_{PD}(t) = \hat{u}(t) + Pe(t) + D\dot{e}(t) \tag{38}$$

where $\hat{u}(t)$ and $u_{PD}(t)$ are the outputs of inverse RBF-ASO and PD controller, respectively.

## 10 . Stability analysis

**Theorem 1:** With the Gaussian basis function defined in (30), the inverse model-based RBF can be written as:

$$u = \sum_{j=0}^{h} w_j \varnothing_j \left( \|x - \mu_j\| \right) \tag{39}$$

It is always upper bounded by $\sum_{j=0}^{h} |w_j|$ and BIBO stable.
**Proof:** The output of the Gaussian basis function of RBF is bounded and can be expressed as follows:

$$0 \le \varnothing_j \left( \|x - \mu_j\| \right) \le 1, j = 0, 1, \ldots, h \tag{40}$$

Assume that

$$|u| > \sum_{j=0}^{h} |w_j| \tag{41}$$

Substituting Equation (39) in Equation (41),

$$\sum_{j=0}^{h} w_j \varnothing_j \left( \|x - \mu_j\| \right) > \sum_{j=0}^{h} |w_j| \tag{42}$$

$$\sum_{j=0}^{h} |w_j| \, \varnothing_j \left( \|x - \mu_j\| \right) > \sum_{j=0}^{h} |w_j| \tag{43}$$

$$\sum_{j=0}^{h} |w_j| \, \varnothing_j \left( \|x - \mu_j\| \right) - \sum_{j=0}^{h} |w_j| > 0 \tag{44}$$

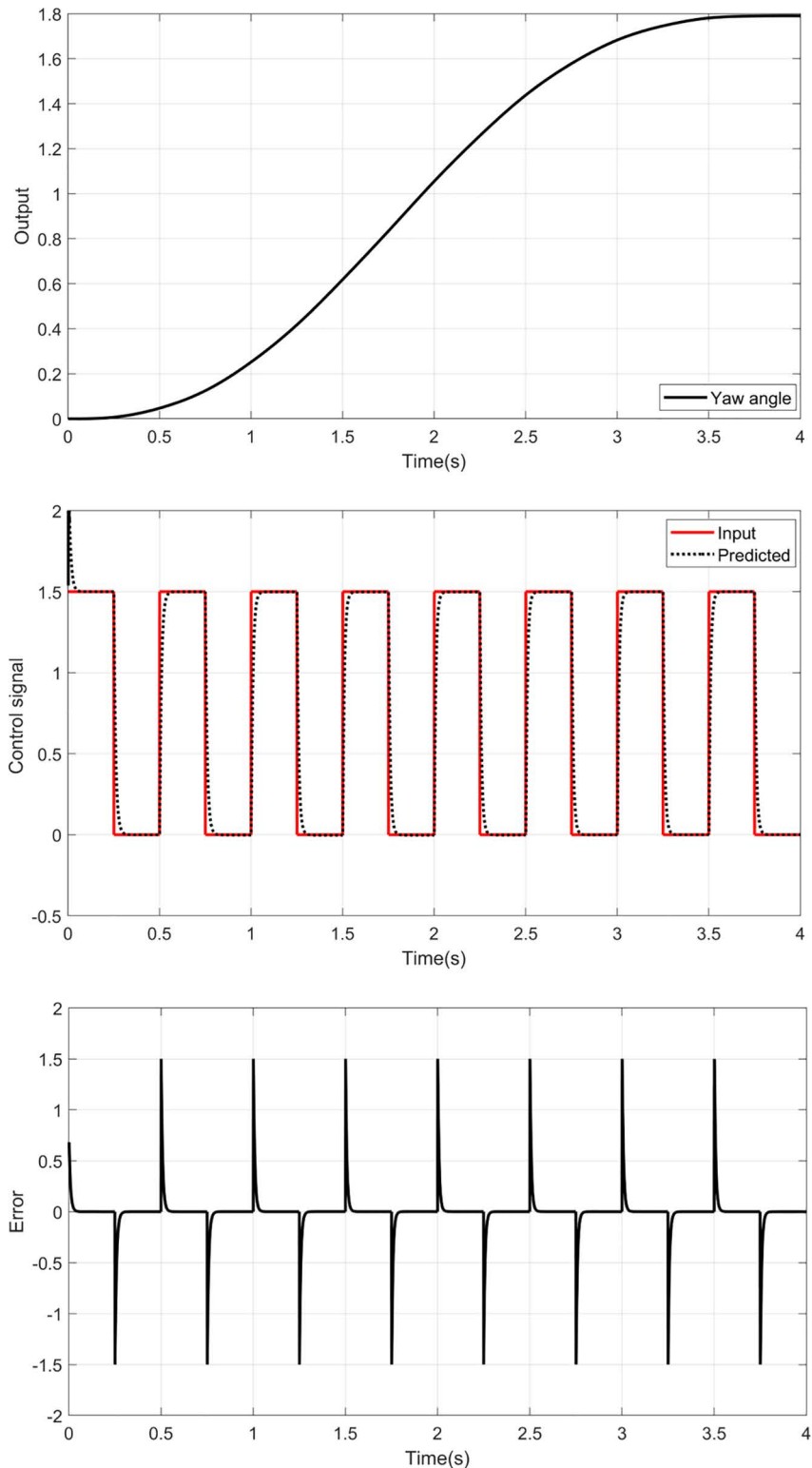

**Fig 5. Training results for Yaw model.**

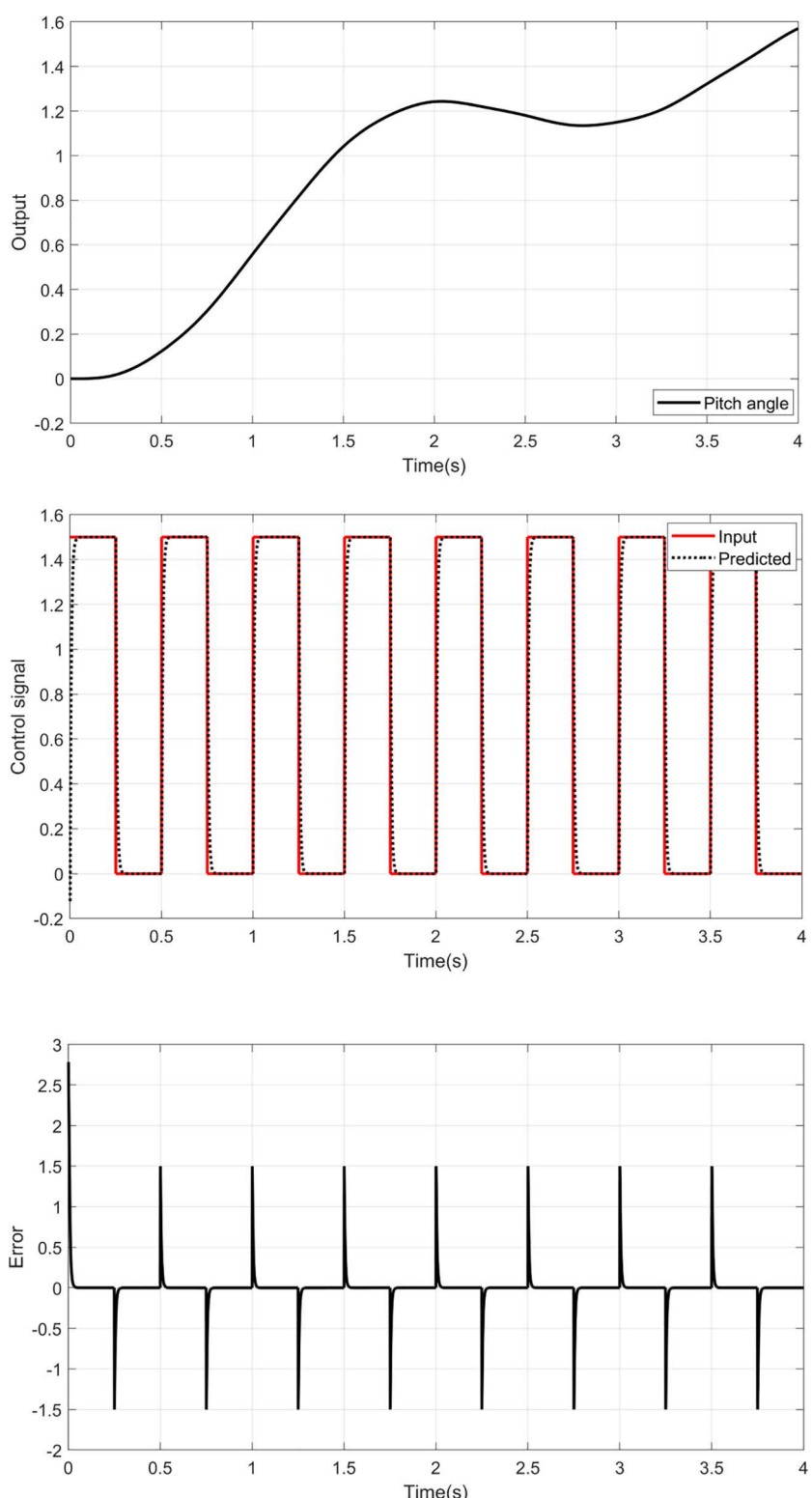

**Fig 6. Training results for Pitch model.**

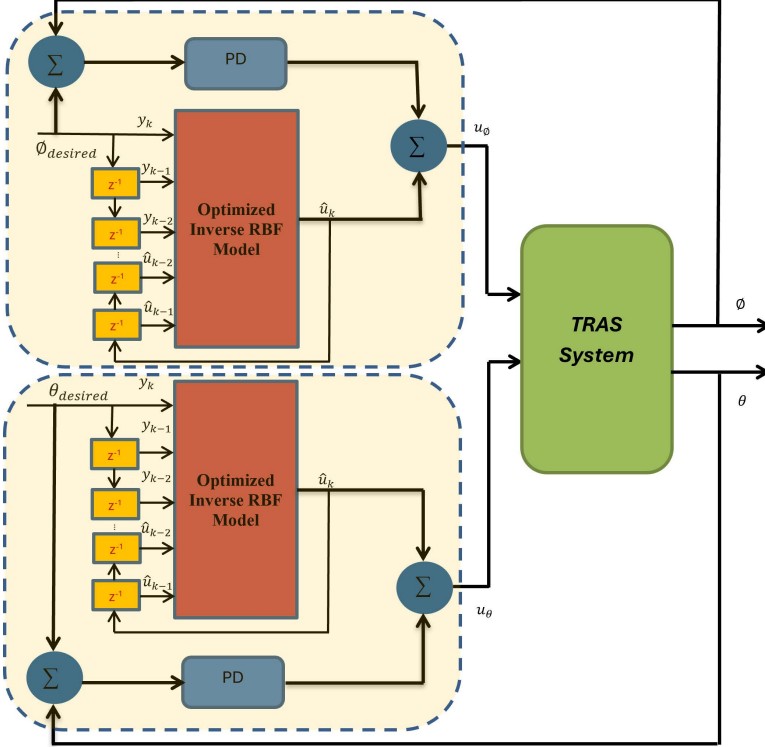

**Fig 7. The proposed control scheme.**

$$\sum_{j=0}^{h} |w_j| \ (\varnothing_j(\|x - \mu_j\|) - 1) > 0 \tag{45}$$

Equation (44) indicates that $\varnothing_j(\|x - \mu_j\|)$ must be greater than 1, while $\varnothing_j(\|x - \mu_j\|)$ must be less than 1 as expressed in Equation (40). Thus, the assumption in Equation (41) is incorrect and the correct one is:

$$|u| < \sum_{j=0}^{h} |w_j| \tag{46}$$

**Theorem 2:** Consider the linearized TRAS system described by Equation (10) with the proposed control law that is defined in Equation (38), the closed loop-controlled system is stable.

**Proof**

Let $V$ be the candidate Lyapunov function, and is expressed as follows:

$$V = V_1 + V_2 \tag{47}$$

Where:

$$V_1 = \frac{1}{2} x_\varnothing^T x_\varnothing \tag{48}$$

$$V_2 = \frac{1}{2} x_\theta^T x_\theta \tag{49}$$

Differentiating Equation (48) gives:

$$\dot{V}_1 = x_\varnothing^T \dot{x}_\varnothing \tag{50}$$

Substituting Equation (10) gives:

$$\dot{V}_1 = x_\varnothing^T [A_\varnothing x_\varnothing + B_\varnothing u_\varnothing] \tag{51}$$

Now, express Equation (51) in terms of Equation (38) as follows:

$$\dot{V}_1 = x_\varnothing^T [A_\varnothing x_\varnothing + B_\varnothing (\hat{u}_{p\varnothing}(t) + u_{PD}(t))] \tag{52}$$

$$= x_\varnothing^T [A_\varnothing x_\varnothing + B_\varnothing (\hat{u}_{p\varnothing}(t) + P_\varnothing e(t) + D_\varnothing \dot{e}(t))] \tag{53}$$

$$= x_\varnothing^T [A_\varnothing x_\varnothing + B_\varnothing (\hat{u}_{p\varnothing}(t) + P_\varnothing (x_{1\varnothing}^d - x_\varnothing(1)) + D_\varnothing (\dot{x}_{1\varnothing}^d - \dot{x}_\varnothing(1)))] \tag{54}$$

$$= x_\varnothing^T [A_\varnothing x_\varnothing + B_\varnothing (\hat{u}_{p\varnothing}(t) + P_\varnothing (x_{1\varnothing}^d - [1\ 0\ 0]x_\varnothing) + D_\varnothing (\dot{x}_{1\varnothing}^d - [0\ 1\ 0]x_\varnothing))] \tag{55}$$

$$= x_\varnothing^T [A_\varnothing x_\varnothing + B_\varnothing (\hat{u}_{p\varnothing}(t) - [P_\varnothing\ 0\ 0]x_\varnothing - [0\ D_\varnothing\ 0]x_\varnothing + P_\varnothing x_{1\varnothing}^d + D_\varnothing \dot{x}_{1\varnothing}^d)] \tag{56}$$

$$= x_\varnothing^T [A_\varnothing x_\varnothing - B_\varnothing ([P_\varnothing\ 0\ 0]x_\varnothing + [0\ D_\varnothing\ 0]x_\varnothing) + B_\varnothing (\hat{u}_{p\varnothing}(t) + P_\varnothing x_{1\varnothing}^d + D_\varnothing \dot{x}_{1\varnothing}^d)] \tag{57}$$

$$= x_\varnothing^T [A_\varnothing - B_\varnothing ([P_\varnothing\ 0\ 0] + [0\ D_\varnothing\ 0])] x_\varnothing + x_\varnothing^T B_\varnothing [\hat{u}_{p\varnothing}(t) + P_\varnothing x_{1\varnothing}^d + D_\varnothing \dot{x}_{1\varnothing}^d] \tag{58}$$

$$= -x_\varnothing^T A_{new} x_\varnothing + x_\varnothing^T B_\varnothing [\hat{u}_{p\varnothing}(t) + P_\varnothing x_{1\varnothing}^d + D_\varnothing \dot{x}_{1\varnothing}^d] \tag{59}$$

$$\leq -x_\varnothing^T A_{new} x_\varnothing + |x_\varnothing| |B_\varnothing| \left[\sum_{j=0}^{h} |w_{\varnothing j}| + P_\varnothing x_{1\varnothing}^d + D_\varnothing \dot{x}_{1\varnothing}^d\right] \tag{60}$$

where:

$$A_{new} = -(A_\varnothing - B_\varnothing ([P_\varnothing\ 0\ 0] + [0\ D_\varnothing\ 0])) \tag{61}$$

According to Theorem 1, it can be assumed that

$$\sum_{j=0}^{h} |w_{\varnothing j}| + P_\varnothing x_{1\varnothing}^d + D_\varnothing \dot{x}_{1\varnothing}^d < k|x_\varnothing|$$

then

$$\dot{V}_1 \leq -x_\varnothing A_{new} x_\varnothing + |x_\varnothing| |B_\varnothing| k |x_\varnothing| \tag{62}$$

If $|A_{new}| \geq k|B_\varnothing|$, then

$$\dot{V}_1 \leq 0 \tag{63}$$

The same procedures can be applied for $V_2$. Then

$$\dot{V} \leq 0 \tag{64}$$

Since $V \geq 0$ and $\dot{V} \leq 0$, the proposed closed-loop controlled system is stable.

## 11. Simulation results

The proposed control method is tested and validated using the MATLAB/Simulink tool to investigate TRAS performance. The ASO parameters are reported in Table 2. The values of the PD controller in the controller are shown in Table 3.

A comparison is conducted with the proposed FOPID controller in [11]. In addition, a PID controller, in which the parameters are tuned by PSO, has been used for comparison. Table 4 lists the parameter values for competitor controllers.

### 11.1. Reference input tracking

The primary objective of control is minimizing the difference between the axis position of TRAS and the desired position. At first, a step input signal is applied to the TRAS system, and the step responses of the proposed, FOPID and PSO-PID controllers are shown in Figs 8 and 9. The controllers' performances are evaluated in terms of the rise time, settling time, maximum overshoot, Integral Absolute Error (IAE), Integral Time Absolute Error (ITAE), Integral Square Error (ISE) and Integral Time Square Error (ITSE). The figures indicated that the proposed control method has a very small overshoot

**Table 2. The ASO parameters.**

| Parameter | Value |
|-----------|-------|
| $J_A$ | 0.0561 |
| $J$ | 0.2168 |
| $J_\theta$ | 0.0559 |

**Table 3. The PD parameters.**

| Parameter | Value | |
|-----------|-------|---|
| | Yaw model | Pitch model |
| $k_p$ | 0.0561 | 3 |
| $k_d$ | 0.0559 | 0.75 |

**Table 4. The FOPID and PID parameters.**

| Parameter | FOPID [9] | | PSO-PID | |
|-----------|-----------|---|---------|---|
| | Yaw model | Pitch model | Yaw model | Pitch model |
| $k_p$ | 0.1396 | 0.183 | 8.887 | 0.594 |
| $k_i$ | 0.0373 | 0.087 | 5.220 | 0.198 |
| $k_d$ | 0.0413 | 0.040 | 3.344 | -0.803 |
| $\lambda$ | 1.1688 | 1.216 | | |
| $\mu$ | 1.9984 | 1.858 | | |

with a shorter settling time. The control signal of all controllers for both yaw and pitch models are shown in Figs 10 and 11, respectively. Table 5 presents the transient specifications and integral performance indices of TRAS for each controller. The proposed controller exhibits an 88.3% faster rise time, a 96.0% faster settling time, and a 93.8% lower overshoot for the Yaw model, along with a 42.8% faster rise time, a 73.9% faster settling time, and an 86.8% lower overshoot for the Pitch model compared to the FOPID controller. In comparison to the PSO-PID controller, the proposed controller shows a 36.2% faster rise time, an 86.7% faster settling time, and a 59.7% lower overshoot for the Yaw model, as well as a 58.4%

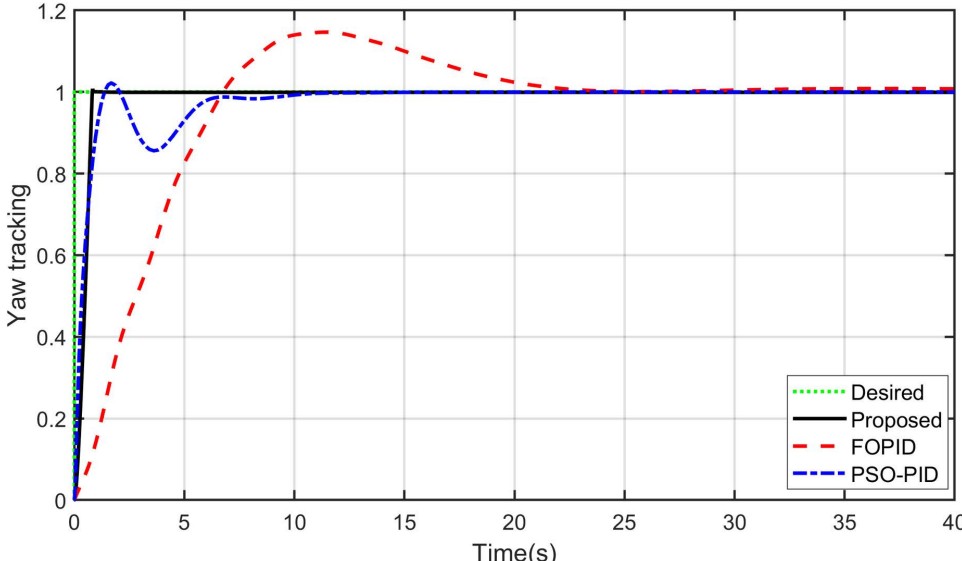

**Fig 8. Step input response of Yaw model.**

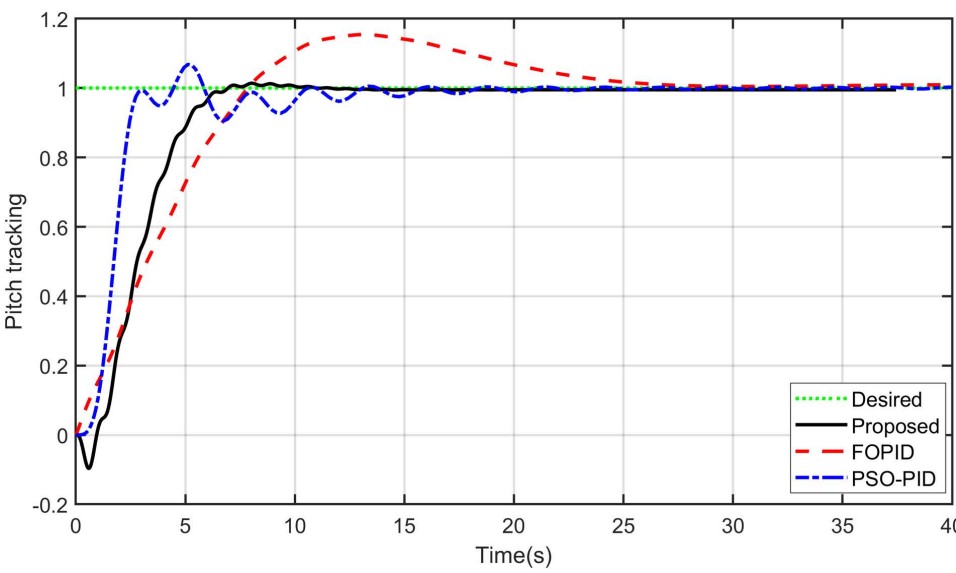

**Fig 9. Step input response of Pitch model.**

slower rise time, a 59.9% faster settling time, and a 71.2% lower overshoot for the Pitch model. These results show that the proposed controller, due to the adaptive nature of the RBF network, demonstrates faster adaptation, as evidenced by its lower rise and settling times. Furthermore, there are significant reductions in the integral performance indices (e.g., IAE, ITAE, ISE and ITSE) for the proposed controller compared to the FOPID and PSO-PID controllers. This indicates that the proposed controller achieves lower tracking errors by dynamically compensating for system nonlinearities. For example, compared to FOPID and PSO-PID, the proposed algorithm reduces the IAE performance index by 93.0% and 41.3%

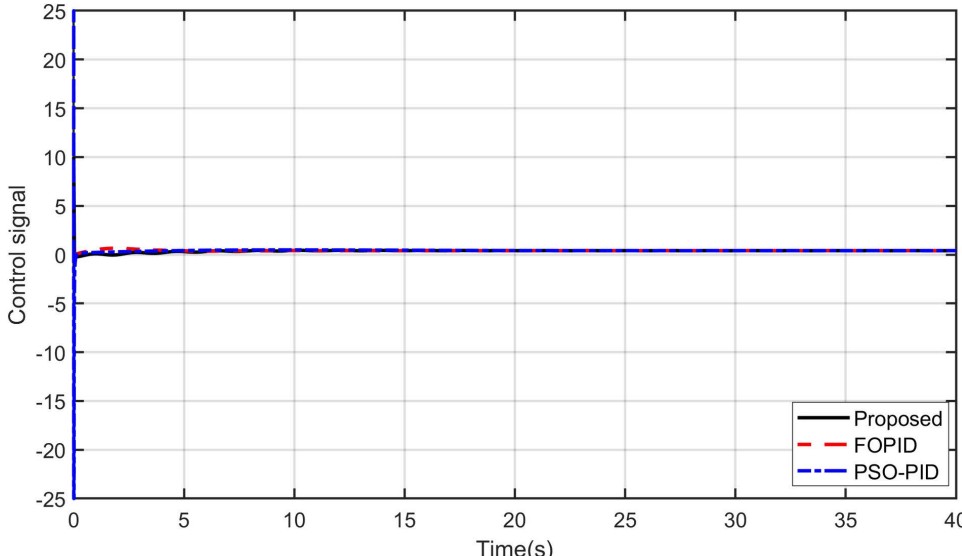

**Fig 10. Control signal for Yaw model.**

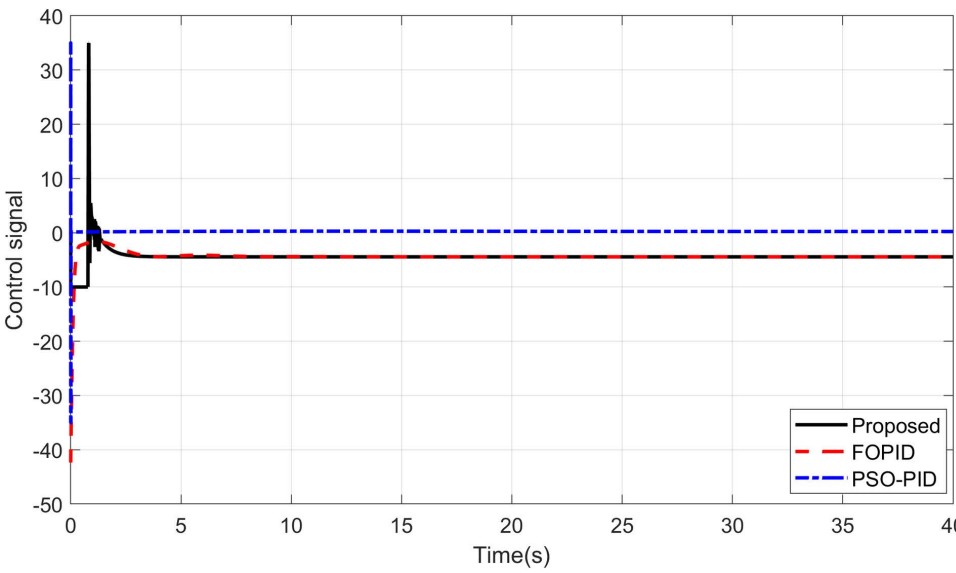

**Fig 11. Control signal for Pitch model.**

for the Yaw model, and by 97.7% and 94.7% for the Pitch model, respectively. Overall, the proposed controller demonstrates noticeable improvements over both FOPID and PSO-PID.

Furthermore, it should be noted that the proposed controller is able to track other reference signals. A sinusoidal input is applied to the yaw model and a square signal is applied to the pitch model. The responses of the controllers to these signals are shown in Figs 12 and 13. As observed, it can be concluded that the proposed controller tracks sinusoidal input to the yaw model more accurately than others do. Concerning the pitch model, the proposed controller tracks the square input signal better among the other controllers. The control signals are shown in Figs 14 and 15. After analyzing the results, it can be stated that the proposed scheme is better than the PID controller in terms of tracking different desired input signals.

## 11.2. Robustness analysis

Robustness is an essential feature of strong and effective controllers. Thus, the proposed controller is compared against the PID controller in terms of robustness and performance. A pulse disturbance signal, shown in Fig 16, with an amplitude

**Table 5. Performance indexes.**

| Index | Proposed controller | | FOPID [9] | | PSO-PID | |
|---|---|---|---|---|---|---|
| | Yaw model | Pitch model | Yaw model | Pitch model | Yaw model | Pitch model |
| Rise Time | 0.5856 | 3.5160 | 5.0185 | 6.1516 | 0.9178 | **1.4567** |
| Settling Time TimeTime | 0.8037 | **6.0845** | 20.0606 | 23.2988 | 6.0473 | 15.1716 |
| Max. Overshoot | 0.8911 | **1.8953** | 14.2978 | 14.3459 | 2.2120 | 6.5729 |
| IAE | 0.5111 | **0.1175** | 7.2576 | 5.1713 | 0.8703 | 2.2156 |
| ITAE | 0.8701 | **3.3480** | 31.5345 | 14.0580 | 2.1370 | 6.1104 |
| ISE | 0.3527 | **1.1542** | 3.5640 | 2.4392 | 1.2800 | 1.4394 |
| ITSE | 0.0800 | **0.9635** | 13.0758 | 6.6966 | 0.2093 | 1.2727 |

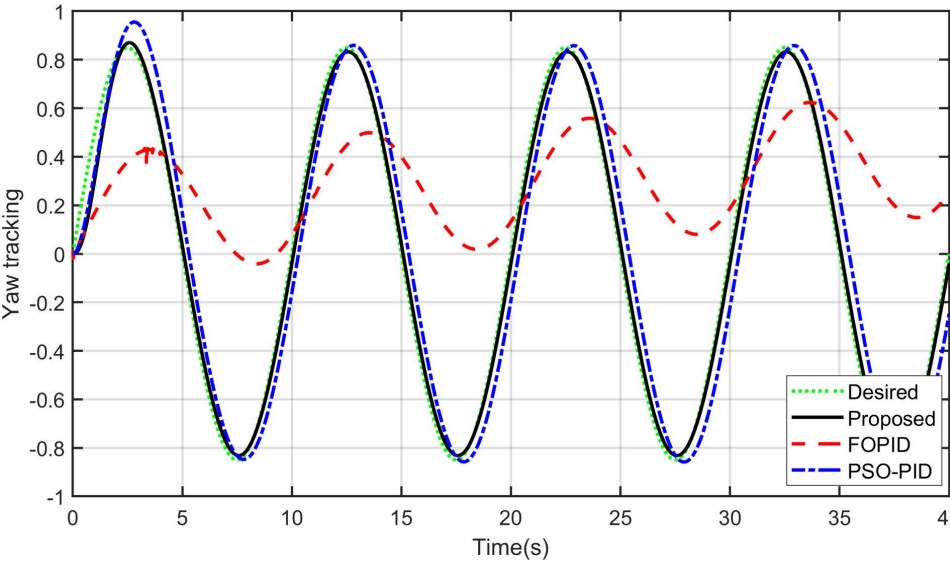

**Fig 12. Sinusoidal tracking of Yaw model.**

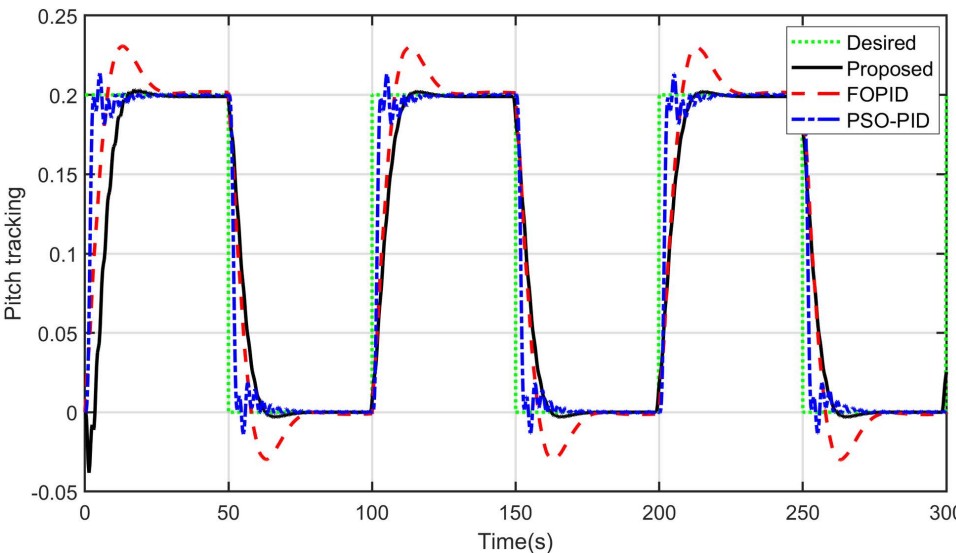

**Fig. 13. Square tracking of Pitch model.**

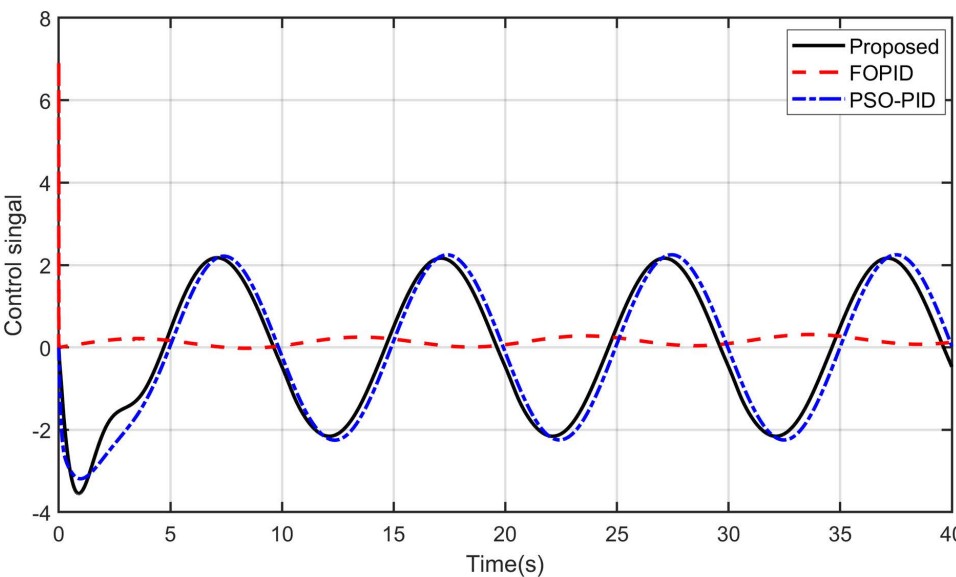

**Fig 14. Control signal for Yaw model.**

of 0.05 and 2-sec width is applied to disturb the yaw and pitch potions. Thus, a comparison between the proposed controller and other controllers was made in terms of disturbance rejection. As shown in Figs 17 and 18, all controllers rejected the disturbance for both the yaw and pitch models. However, the proposed controller is less affected and demonstrates better stability and robustness against disturbances. It requires less time than FOPID and PSO-PID controllers to regain its position. This is due to the hybrid approach's ability to handle high system nonlinearities and maintain stability under challenging conditions.

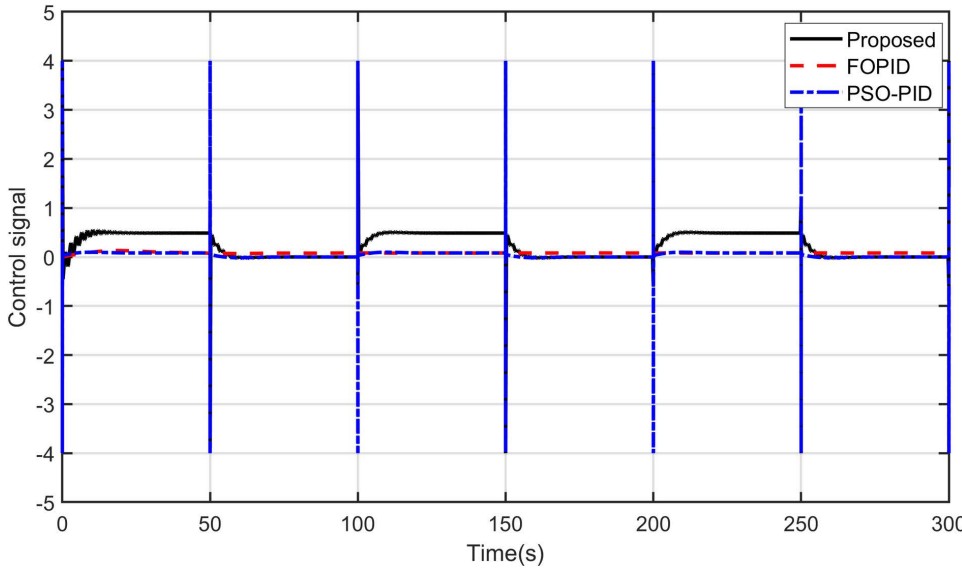

**Fig 15. Control signal for Pitch model.**

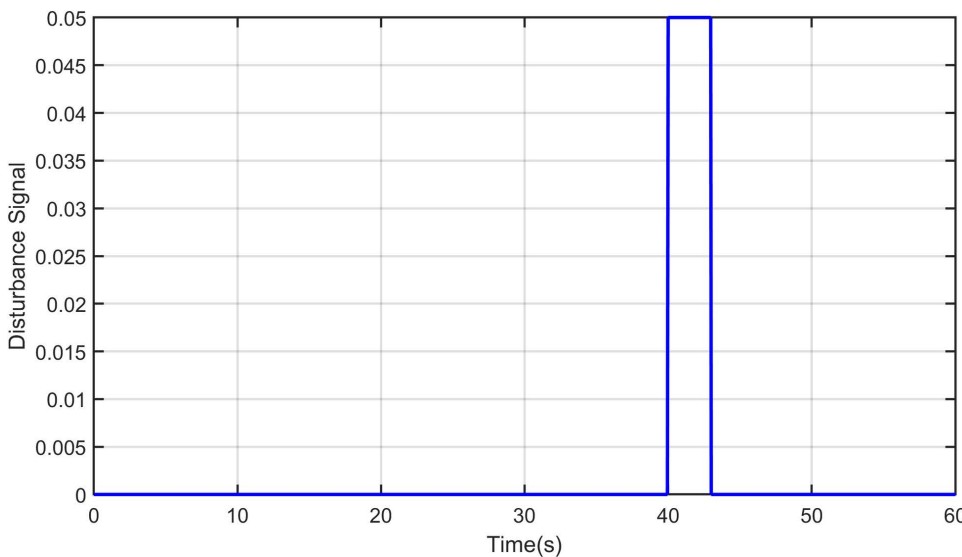

**Fig 16. Disturbance signal applied on Yaw and Pitch model.**

## 12. Conclusions

In this paper, an inverse RBF neural networks model-based ASO identification for TRAS was presented by combining a feed-forward controller with a feedback controller. First, a nonlinear dynamic of TRAS was linearized and decoupled into yaw and pitch models. Inverse models for both yaw and pitch models have been developed according to the approximation capability of the RBF neural networks. To reduce the inverse modeling error, a feedback PD controller has been

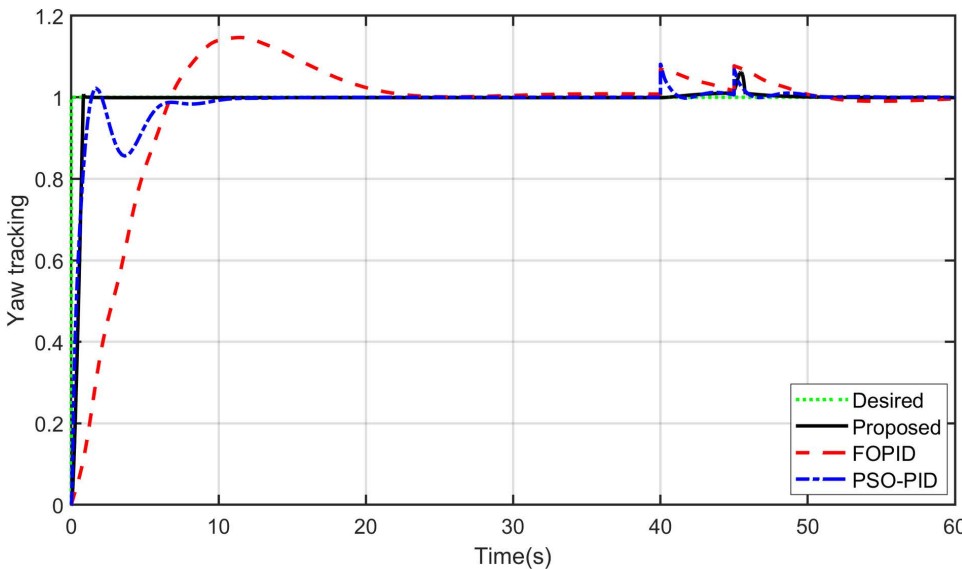

**Fig 17. Disturbance rejection for Yaw model.**

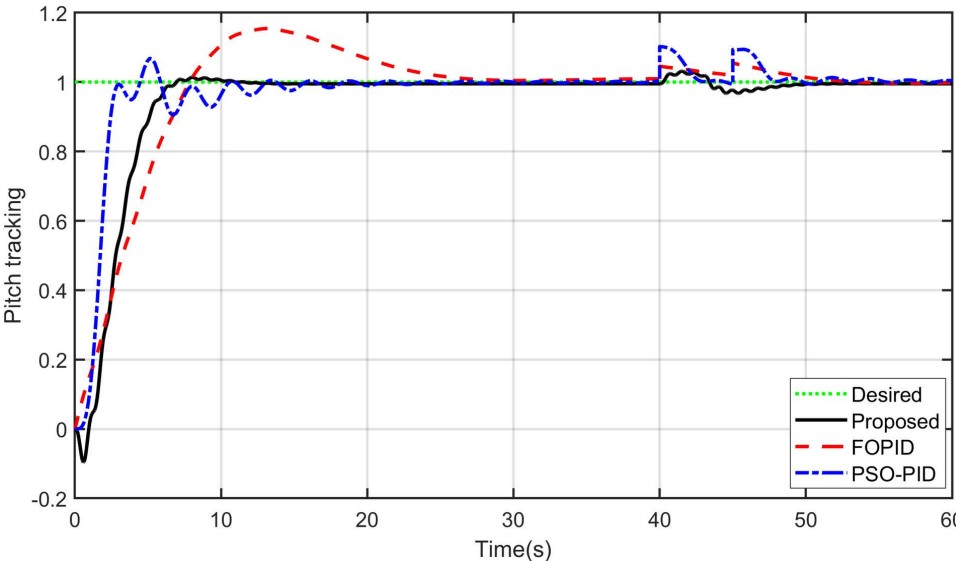

**Fig 18. Disturbance rejection for Pitch model.**

designed and integrated with the fed forward inverse neural controller. A comparison is made between the proposed controller and optimized controllers including FOPID and PID. The simulation results demonstrated that the proposed controller exhibited a good performance in terms of transient specifications and integral time performance indices. Finally, the simulation environment was designed to closely mimic real-world scenarios, including system uncertainties and disturbances, ensuring the robustness and adaptability of the proposed method. Future work could extend the current study to include real-time implementation, which is important for practical validation. In this context, the ASO algorithm and RBF

neural network are lightweight and can be implemented on embedded systems. Additionally, the proposed approach can be integrated with standard control hardware, such as microcontrollers or FPGAs, for real-time deployment.

## Author contributions

**Conceptualization:** Ahmad Al-Talabi.

**Data curation:** Ahmad Al-Talabi, Ali Hussien Mary.

**Formal analysis:** Ahmad Al-Talabi, Aqeel Abdulazeez Mohammed, Ali Hussien Mary.

**Funding acquisition:** Ahmad Al-Talabi.

**Investigation:** Ahmad Al-Talabi, Aqeel Abdulazeez Mohammed, Ali Hussien Mary.

**Methodology:** Ahmad Al-Talabi, Taqwa Oday Fahad, Ali Hussien Mary.

**Project administration:** Ahmad Al-Talabi.

**Resources:** Ahmad Al-Talabi.

**Software:** Ahmad Al-Talabi, Taqwa Oday Fahad.

**Supervision:** Ahmad Al-Talabi, Ali Hussien Mary.

**Validation:** Ahmad Al-Talabi, Aqeel Abdulazeez Mohammed.

**Visualization:** Ahmad Al-Talabi, Ali Hussien Mary.

**Writing – original draft:** Ahmad Al-Talabi.

**Writing – review & editing:** Ahmad Al-Talabi.

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
