## [Decision Letter · Decision Letter 0]

30 Dec 2024

PONE-D-24-45546Inverse Modeling, Analysis and Control of Twin Rotor Aerodynamic Systems with Optimized Artificial Intelligent ControllersPLOS ONE

Dear Dr. Al-Talabi,

Thank you for submitting your manuscript to PLOS ONE. After careful consideration, we feel that it has merit but does not fully meet PLOS ONE’s publication criteria as it currently stands. Therefore, we invite you to submit a revised version of the manuscript that addresses the points raised during the review process.

**ACADEMIC EDITOR: **

The manuscript presents a novel approach to controlling Twin-Rotor Aerodynamic Systems (TRAS) using an optimal inverse Radial Basis Function (RBF) neural network. It addresses challenges associated with TRAS's high nonlinearity and coupling. While the study is innovative, it lacks empirical validation, and several areas require clarity and presentation enhancement.

**Key Points for Revision:**

Include real-time implementation or a detailed discussion on mapping simulation results to practical systems.Provide a more rigorous error analysis using additional metrics (e.g., ITAE, ISE) and comparative studies with traditional methods.Update the literature review with relevant recent works and explicitly highlight the novelty of the proposed hybrid controller.Improve the quality of figures, standardize captions, and enhance table formats for clarity.Strengthen the proof of Theorem 2 by elaborating transitions and justifying key inequalities.Address system uncertainties and robustness in real-time applications supported by empirical or case-study evidence.Thoroughly proofread for typos, standardize terminology (TRAS/TRMS), and ensure consistent notation.

These revisions are critical to ensure the manuscript meets the clarity, rigor, and practical applicability expected in this field. We encourage the authors to address these points thoroughly for consideration in the following review cycle.

We look forward to receiving your revised manuscript.

Kind regards,

Kranthi Kumar Deveerasetty, Ph.D.

Academic Editor

PLOS ONE

Journal Requirements:

4. Please remove your figures from within your manuscript file, leaving only the individual TIFF/EPS image files, uploaded separately. These will be automatically included in the reviewers’ PDF.

Additional Editor Comments:

The manuscript presents a novel approach to controlling Twin-Rotor Aerodynamic Systems (TRAS) using an optimal inverse Radial Basis Function (RBF) neural network. It addresses challenges associated with TRAS's high nonlinearity and coupling. While the study is innovative, it lacks empirical validation, and several areas require clarity and presentation enhancement.

Key Points for Revision:

Include real-time implementation or a detailed discussion on mapping simulation results to practical systems.

Provide a more rigorous error analysis using additional metrics (e.g., ITAE, ISE) and comparative studies with traditional methods.

Update the literature review with relevant recent works and explicitly highlight the novelty of the proposed hybrid controller.

Improve the quality of figures, standardize captions, and enhance table formats for clarity.

Strengthen the proof of Theorem 2 by elaborating transitions and justifying key inequalities.

Address system uncertainties and robustness in real-time applications supported by empirical or case-study evidence.

Thoroughly proofread for typos, standardize terminology (TRAS/TRMS), and ensure consistent notation.

These revisions are critical to ensure the manuscript meets the clarity, rigor, and practical applicability expected in this field. We encourage the authors to address these points thoroughly for consideration in the following review cycle.

Reviewers' comments:

Reviewer's Responses to Questions

**Comments to the Author**

1. Is the manuscript technically sound, and do the data support the conclusions?

Reviewer #1: Partly

Reviewer #2: Partly

2. Has the statistical analysis been performed appropriately and rigorously? 

Reviewer #1: Yes

Reviewer #2: Yes

3. Have the authors made all data underlying the findings in their manuscript fully available?

Reviewer #1: Yes

Reviewer #2: No

4. Is the manuscript presented in an intelligible fashion and written in standard English?

Reviewer #1: Yes

Reviewer #2: Yes

5. Review Comments to the Author

Reviewer #1: - Include experimental results on a TRAS since at the moment, the work is limited to the simulation environment only. At least, please include a discussion on, to what extent, are the simulation results can be mapped to control a real system?

- More rigorous analysis of error is required e.g. include results on other parameters such as ITAE, ISE, etc.

- It is mentioned that a TRAS can find potential to be used as a test bed for analysing the performance of control algorithms for helicopters, UAVs, etc. Include related studies exploiting this feature of TRAS such as; 'Differentiator- and Observer-Based Feedback Linearized Advanced Nonlinear Control Strategies for an Unmanned Aerial Vehicle System' and 'Adaptive optimal control of twin-rotor helicopter system; Meta-heuristics and Lyapunov based approach'.

- Include the Paper Outlines at the end of Section 1 (Introduction).

- Both words are used in the paper; TRAS and TRMS. I suppose they are same? If yes, please make them consistent.

- PD controller is used. What is the rationale for that? because the Integral component could have helped to further improve the Steady State error.

- Update your literature review on control of TRAS including notable references such as 'A flexible mixed-optimization with H∞ control for coupled twin rotor MIMO system based on the method of inequality (MOI)- An Experimental Study'.

- Include the stability analysis of the proposed hybrid control law.

- The simplicity feature of a PID-based control law, mentioned in Section 1 (Introduction) could benefit from the reference 'Embedded control system for autarep - a novel autonomous articulated robotic educational platform'.

- At the end of the Abstract, please include 1-2 sentences to quantitatively summarise the key findings of your proposed hybrid control algorithm. 

- It is mentioned that FOPID is better and smoother than the traditional PID controllers. This statement could be crisper if supported with a literature work such as 'Robust MPPT control of stand-alone photovoltaic systems via adaptive fractional-order PID controller with self-adjusting fractional orders'.

- Does the novelty of the proposed work lie in proposing a hybrid controller (combination of PD with inverse RBF)? Please explicitly mention the novelty element.

- In Table 1, include a new column 'Description' where each parameter is briefly mentioned what it is? Also, the units may be moved to a new column.

- In Figure 4, include the signs to each feedback /feed-forward loop (the is the inputs to each summer should be assigned a sign). I suppose it is positive?

- In Table 2, write 0 with .0561

- Please improve the quality of all Figures.

- Also, thoroughly proofread the paper for typos and other linguistic improvements.

Reviewer #2: This paper presents a novel approach to controlling Twin Rotor Aerodynamic Systems (TRAS) using an optimal inverse Radial Basis Function (RBF) neural network model, addressing the challenges posed by the high nonlinearity and coupling effects in TRAS, modeled as Multi-Input Multi-Output (MIMO) systems. The paper is well-written, but a few improvements are needed to enhance clarity and rigor. Below are my detailed comments:

Major Remarks:

Real-Time Implementation and Robustness: Could you provide more details on how the proposed approach (ASO with inverse RBF neural networks) improves traditional methods in practical scenarios? Specifically, how does it perform in real-time implementation and under system uncertainties? Empirical evaluation or case studies on these aspects would strengthen the paper's practical relevance.

Figure Captions Consistency: The captions in Figures 8, 10, 14, and 16 are not uniform. It is recommended to standardize the phrasing and style to ensure consistency and improve the overall readability and professionalism of the presentation.

Clarification of Theorem 2 Proof: The proof of Theorem 2 is well-structured, but the transition between equations (50) to (54) lacks clarity. Provide explicit reasoning for substitutions such as $ \hat{u}_p(t) $, $ P_{\phi} e(t) $, and $ D_{\phi} \dot{e}(t) $ would be helpful. Additionally, clarify the summation $ \sum |w_j| $ and its relationship to $ x_{\phi} $ in Equation (55), justify the inequality $ |A_{\text{new}}| \leq k |B_{\phi}| $ in Equation (56), and elaborate on how $ \dot{V} \leq 0 $ ensures stability via the Lyapunov criterion would strengthen the rigor of the proof.

Comparison with Traditional Methods: Could you elaborate on why the proposed approach outperforms traditional PID and hybrid methods ([11]-[14])? Specifically, how does the inverse RBF neural network, optimized using the ASO algorithm, better handle high system nonlinearities? A clear comparison of performance metrics like tracking accuracy, adaptability, and robustness would highlight the advantages of the proposed method.

Minor Remarks:

There is a typo in the constant after Equation 20. Please ensure that the constant ‘kbest’ is corrected for consistency throughout the paper.

Correct MATLAB notation in Section 10: “Simulation Results”.

Please correct the notation for the sample (n) to ensure consistency throughout the document wherever required.

6. PLOS authors have the option to publish the peer review history of their article (what does this mean? ). If published, this will include your full peer review and any attached files.

**Do you want your identity to be public for this peer review?** For information about this choice, including consent withdrawal, please see our Privacy Policy .

Reviewer #1: No

Reviewer #2: No

---

## [Author Response · Author response to Decision Letter 1]

6 Mar 2025

First of all, we would like to thank the reviewers for their valuable comments and suggestions. Please find our detailed responses in blue.

Reviews **********************

Reviewer 1:

1. Include experimental results on a TRAS since at the moment, the work is limited to the simulation environment only. At least, please include a discussion on, to what extent, are the simulation results can be mapped to control a real system?

Authors response: Thanks to the reviewer for his valuable feedback. The authors appreciate his suggestion to include experimental results on TRAS to provide a solid foundation for the work in this field. The results presented in this manuscript serve as a strong theoretical basis for the effectiveness of the proposed approach. Additionally, the simulation environment was designed to closely mimic real-world scenarios, including system uncertainties and disturbances, to ensure the robustness and adaptability of the proposed method. While real-time implementation is beyond the scope of this paper, we acknowledge its importance for practical validation. The ASO algorithm and RBF neural network are lightweight and can be implemented on embedded systems. Moreover, the proposed method can be integrated with standard control hardware, such as microcontrollers or FPGAs, for real-time deployment. The authors look forward to incorporating these enhancements into their future work as illustrated in Section 12 of the revised manuscript (please refer to page 25).

2. More rigorous analysis of error is required e.g. include results on other parameters such as ITAE, ISE, etc.

Authors response: Thanks to the Reviewer for his insightful suggestion to include a more rigorous analysis of error using performance indices such as ITAE, ISE and ITSE. We agree that this will strengthen the robustness of our results. In response to the reviewer's suggestion, we have calculated the ITAE, ISE and ITSE for each controller used in the simulation. These metrics were evaluated and are now included in the revised manuscript, as showed in Table 5 (please refer to page 19 of the revised manuscript).

3. It is mentioned that a TRAS can find potential to be used as a test bed for analysing the performance of control algorithms for helicopters, UAVs, etc. Include related studies exploiting this feature of TRAS such as; 'Differentiator- and Observer-Based Feedback Linearized Advanced Nonlinear Control Strategies for an Unmanned Aerial Vehicle System' and 'Adaptive optimal control of twin-rotor helicopter system; Meta-heuristics and Lyapunov based approach'.

Authors response: Thanks to the Reviewer for his valuable suggestion to include further related studies that highlight the use of the TRAS as a test bed for analyzing control algorithms for helicopters, UAVs, and similar systems. We agree that this will support our work. Thus, following the reviewer's recommendation, this issue is considered in the revised manuscript. Specifically, we have added the following studies to the revised manuscript:

'Differentiator- and Observer-Based Feedback Linearized Advanced Nonlinear Control Strategies for an Unmanned Aerial Vehicle System'

'Adaptive optimal control of twin-rotor helicopter system; Meta-heuristics and Lyapunov based approach'

These studies have been cited in the Introduction Section and added to the Reference Section (please refer to pages 2 and 25 of the revised manuscript).

4. Include the Paper Outlines at the end of Section 1 (Introduction).

Authors response: Thanks to the Reviewer for noting this. The Reviewer’s comment alerted us that the Paper Outlines are missing. This issue is addressed in the revised manuscript by adding the Paper outlines at the end of the Introduction section (please refer to page 3 of the revised manuscript).

5. Both words are used in the paper; TRAS and TRMS. I suppose they are same? If yes, please make them consistent.

Authors response:

Thanks to the Reviewer for his comment. The Reviewer is correct in noting that both two term TRAS and TRMS are used within the manuscript, which are the same. Therefore, the authors modified the manuscript to only used the acronym TRAS (please refer to page 16 of the revised manuscript).

6. PD controller is used. What is the rationale for that? because the Integral component could have helped to further improve the Steady State error.

Authors response: Thanks to the reviewer for his valuable comment. In response, to explain the rationale for using a PD controller instead of a PID controller, we need to demonstrate why the integral component was excluded despite its capability to improve steady-state error, and how the PD controller's design achieves our goal. The decision was based on the following considerations:

Our design focuses on fast response, stability and minimizing steady state error.

As we know, the Proportional part ensures a fast response, while the Derivative component improves stability by damping oscillations and reducing overshoot. Therefore, the PD controller makes the system faster with minimal overshoot.

While the Integral part is traditionally used to eliminate steady-state error, it can lead to integral windup in systems with high nonlinearities or saturation limits, causing instability or excessive overshoot.

In Section 10, we prove the stability of the proposed controller using only a PD controller, which simplifies our system compared to a PID controller.

The RBF network dynamically adjusts its parameters to compensate for system nonlinearities and disturbances which effectively reduces steady-state error.

The ASO algorithm ensures that the controller's parameters are optimally tuned, further minimizing steady-state error.

In conclusion, in our proposed approach, the inverse RBF neural network, optimized by ASO, ensures that steady-state error is minimized without the need for an Integral part.

7. Update your literature review on control of TRAS including notable references such as 'A flexible mixed-optimization with H∞ control for coupled twin rotor MIMO system based on the method of inequality (MOI) - An Experimental Study'.

Authors response: Thanks to the Reviewer for his valuable suggestion to update the literature review on the control of TRAS and we appreciate his recommendation to include the notable reference:

'A flexible mixed-optimization with H∞ control for coupled twin rotor MIMO system based on the method of inequality (MOI) - An Experimental Study'.

We agree that this reference is highly relevant to our work and will strengthen the context of our literature review. It has been cited in the Introduction Section of the revised manuscript, and its findings have been highlighted. It is added to the Reference Section (please refer to pages 2 and 26 of the revised manuscript).

8. Include the stability analysis of the proposed hybrid control law.

Authors response: Thanks to the Reviewer for his comment. Section 10 of the revised manuscript provides the closed-loop stability analysis of the proposed hybrid control law using the Lyapunov stability theorem (please refer to pages 16-17 of the revised manuscript).

9. The simplicity feature of a PID-based control law, mentioned in Section 1 (Introduction) could benefit from the reference 'Embedded control system for autarep - a novel autonomous articulated robotic educational platform'.

Authors response: Thanks to the Reviewer for the valuable suggestion to include the notable reference:

"Embedded control system for autarep - a novel autonomous articulated robotic educational platform".

We agree that incorporating this reference will further support and strengthen the discussion on the simplicity of the PID-based control law. In response to the reviewer's recommendation, we have revised the Introduction and Reference Sections to include this reference (please refer to pages 2 and 26 of the revised manuscript).

10. At the end of the Abstract, please include 1-2 sentences to quantitatively summarise the key findings of your proposed hybrid control algorithm.

Authors response: Thanks to the reviewer for his valuable comment. Following the Reviewer's advice, we have enhanced the abstract by adding, at the end, a quantitative summary of the key findings of our proposed hybrid control algorithm (please refer to page 1 of the revised manuscript).

11. It is mentioned that FOPID is better and smoother than the traditional PID controllers. This statement could be crisper if supported with a literature work such as 'Robust MPPT control of stand-alone photovoltaic systems via adaptive fractional-order PID controller with self-adjusting fractional orders'.

Authors response: We appreciate the reviewer's insightful suggestion to strengthen the statement regarding the superiority of FOPID over traditional PID controllers. In response to this suggestion, we have revised the manuscript and included the suggested reference (please refer to pages 2 and 26 of the revised manuscript).

'Robust MPPT control of stand-alone photovoltaic systems via adaptive fractional-order PID controller with self-adjusting fractional orders'

Which provides evidence from the literature to support our assertion that FOPID offers better and smoother performance compared to traditional PID controllers.

12. Does the novelty of the proposed work lie in proposing a hybrid controller (combination of PD with inverse RBF)? Please explicitly mention the novelty element.

Authors response: Thanks to the Reviewer for his important comment regarding the novelty of our proposed work. The Reviewer is correct in noting that the novelty of our work lies in proposing a hybrid controller that combines a Proportional-Derivative (PD) controller with an inverse Radial Basis Function (RBF) neural network model-based ASO identification for TRAS. This hybrid approach exploits the stability and simplicity of the PD controller while utilizing the adaptive and learning capabilities of the inverse RBF to handle system nonlinearities and uncertainties. To the best of our knowledge, this specific combination has not been extensively explored in the context of TRAS (please refer to pages 2-3 of the revised manuscript, where a clear statement is provided).

13. In Table 1, include a new column 'Description' where each parameter is briefly mentioned what it is? Also, the units may be moved to a new column.

Authors response: We appreciate the Reviewer's suggestion to enhance the clarity and organization of Table 1. In response, we have added a 'Description' column and separated the units into a new column, as illustrated in Table 1 on page 5 of the revised manuscript. The updated table now provides a clearer and more detailed presentation of the parameters, their descriptions, and their respective units.

14. In Figure 4, include the signs to each feedback /feed-forward loop (the is the inputs to each summer should be assigned a sign). I suppose it is positive?

Authors response: Thanks to the Reviewer for noting this. The reviewer's comment alerted us to that the signs of the feedback and feed-forward loops were missing. In response, we have addressed this issue by revising Figure 4 to include the signs for each input to the summing junctions (please refer to Figure 4 on page 12 of the revised manuscript).

15. In Table 2, write 0 with .0561

Authors response: Thanks to the Reviewer for his attention to detail. The authors have made the required change (please refer to Table 2 on page 15 of the revised manuscript).

16. Please improve the quality of all Figures.

Authors response: Thanks to the Reviewer for the valuable suggestion to improve the quality of the figures in our manuscript. Following the Reviewer’s recommendation, all Figures have been reproduced in the revised manuscript to ensure better quality.

17. Also, thoroughly proofread the paper for typos and other linguistic improvements.

Authors response: To address the Reviewer’s concern, all the textual corrections have been made, and the manuscript was proofread by a native English speaker from Canada. Also, the manuscript has been subjected to thorough revisions, and the typographical and grammatic issues have been essentially eliminated.

Reviewer 2:

This paper presents a novel approach to controlling Twin Rotor Aerodynamic Systems (TRAS) using an optimal inverse Radial Basis Function (RBF) neural network model, addressing the challenges posed by the high nonlinearity and coupling effects in TRAS, modeled as Multi-Input Multi-Output (MIMO) systems. The paper is well-written, but a few improvements are needed to enhance clarity and rigor. Below are my detailed comments:

Major Remarks:

1. Real-Time Implementation and Robustness: Could you provide more details on how the proposed approach (ASO with inverse RBF neural networks) improves traditional methods in practical scenarios? Specifically, how does it perform in real-time implementation and under system uncertainties? Empirical evaluation or case studies on these aspects would strengthen the paper's practical relevance.

Authors response: We appreciate the reviewer's input. In response, we would like to explain that a similar comment was raised by the first Reviewer, and the answer to this comment is provided in our first response.

2. Figure Captions Consistency: The captions in Figures 8, 10, 14, and 16 are not uniform. It is recommended to standardize the phrasing and style to ensure consistency and improve the overall readability and professionalism of the presentation.

Authors response: Thanks to the reviewer for pointing out the inconsistency in the captions of Figures 8, 10, 14, and 16. We have revised the captions to ensure uniformity in phrasing and style, improving readability and professionalism (please refer to pages 20, 21, 23-24 of the revised manuscript).

3. Clarification of Theorem 2 Proof: The proof of Theorem 2 is well-structured, but the transition between equations (50) to (54) lacks clarity. Provide explicit reasoning for substitutions such as $ \hat{u}_p(t) $, $ P_{\phi} e(t) $, and $ D_{\phi} \dot{e}(t) $ would be helpful. Additionally, clarify the summation $ \sum |w_j| $ and its relationship to $ x_{\phi} $ in Equation (55), justify the inequality $ |A_{\text{new}}| \leq k |B_{\phi}| $ in Equation (56), and elaborate on how $ \dot{V} \leq 0 $ ensures stability via the Lyapunov criterion would strengthen the rigor of the proof.

Authors response: Thanks to the reviewer for his valuable comment.

Following the reviewer's recommendation, we have enhanced the technical understanding of Equations (50) to (54) by providing more details on its mathematical formulation. The equations are clearly presented and defined. Additionally, the relationship between the equations is provided. The details mathematical transition between equations (50) to (54) is illustrated by Equations (50) to (59).

Moreover, the inequality has been justified.

The relationship between w_(∅j) and x_∅: The relationship comes from 51, 38 and 39.

4. Comparison with Traditional Methods: Could you elaborate on why the proposed approach outperforms traditional PID and hybrid methods ([11]-[14])? Specifically, how does the inverse RBF neural network, optimized using the ASO algorithm, better handle high system nonlinearities? A clear comparison of performance metrics like tracking accuracy, adaptability, and robustness would highlight the advantages of the proposed method.

Authors response: Thanks to the Reviewer for his important comments, which alerted us that more discussion is required regarding:

The ability of the inverse RBF neural network optimized by ASO to handle high system nonlinearities.

The inverse RBF neural network represents the main component of the proposed controller. Unlike traditional PID controllers, which rely on fixed gains and linear control laws, the inverse RBF neural network can adapt to system nonlinearities. Specifically, it can approximate complex nonlinear functions, allowing it to model and compensate for high system nonlinearities, like TRAS, that traditional PID controllers cannot effectively handle. The network has the ability to learn and adjust its parameters to ensure optimal

---

## [Decision Letter · Decision Letter 1]

1 Apr 2025

Inverse Modeling, Analysis and Control of Twin Rotor Aerodynamic Systems with Optimized Artificial Intelligent Controllers

PONE-D-24-45546R1

Dear Dr. Ahmad,

We’re pleased to inform you that your manuscript has been judged scientifically suitable for publication and will be formally accepted for publication once it meets all outstanding technical requirements.

Kind regards,

Kranthi Kumar Deveerasetty, Ph.D.

Academic Editor

PLOS ONE

Additional Editor Comments (optional):

All comments have been addressed

Reviewers' comments:

Reviewer's Responses to Questions

**Comments to the Author**

1. If the authors have adequately addressed your comments raised in a previous round of review and you feel that this manuscript is now acceptable for publication, you may indicate that here to bypass the “Comments to the Author” section, enter your conflict of interest statement in the “Confidential to Editor” section, and submit your "Accept" recommendation.

Reviewer #1: All comments have been addressed

Reviewer #2: All comments have been addressed

2. Is the manuscript technically sound, and do the data support the conclusions?

Reviewer #1: Yes

Reviewer #2: Yes

3. Has the statistical analysis been performed appropriately and rigorously? 

Reviewer #1: I Don't Know

Reviewer #2: N/A

4. Have the authors made all data underlying the findings in their manuscript fully available?

Reviewer #1: Yes

Reviewer #2: Yes

5. Is the manuscript presented in an intelligible fashion and written in standard English?

Reviewer #1: Yes

Reviewer #2: Yes

6. Review Comments to the Author

Reviewer #1: The authors have addressed all the suggested comments. The revised version of the paper has improved significantly and can be accepted.

Reviewer #2: I am pleased to inform you that after review, your manuscript has been accepted for publication. The reviewer has evaluated your work and found it to be a valuable contribution to the field.

We appreciate your contribution and thank you for the opportunity to consider your research. Congratulations!

7. PLOS authors have the option to publish the peer review history of their article (what does this mean? ). If published, this will include your full peer review and any attached files.

**Do you want your identity to be public for this peer review?** For information about this choice, including consent withdrawal, please see our Privacy Policy .

Reviewer #1: No

Reviewer #2: **Yes: ** Nidhi Agarwal

---

## [Editor Report · Acceptance letter]

PONE-D-24-45546R1

PLOS ONE

Dear Dr. Al-Talabi,

I'm pleased to inform you that your manuscript has been deemed suitable for publication in PLOS ONE. Congratulations! Your manuscript is now being handed over to our production team.

Kind regards,

on behalf of

Dr. Kranthi Kumar Deveerasetty

Academic Editor

PLOS ONE